# Inflammation in Coronary Atherosclerosis: Insights into Pathogenesis and Therapeutic Potential of Anti-Inflammatory Drugs

**DOI:** 10.3390/ph16091242

**Published:** 2023-09-01

**Authors:** Clara Salles Figueiredo, Elias Soares Roseira, Tainá Teixeira Viana, Marcelo Augusto Duarte Silveira, Rodrigo Morel Vieira de Melo, Miguel Godeiro Fernandez, Livia Maria Goes Lemos, Luiz Carlos Santana Passos

**Affiliations:** 1Programa de Pós Graduação em Medicina e Saúde (Graduate Program in Medicine and Health), Federal University of Bahia, Salvador 40110-060, BA, Brazillpassos8@gmail.com (L.C.S.P.); 2Hospital Ana Nery, Salvador 40301-155, BA, Brazil; elias.soares@ufba.br (E.S.R.);

**Keywords:** coronary artery disease, anti-inflammatories, myocardial ischemia

## Abstract

Atherosclerosis is a lipid-driven immune-inflammatory disease that affects the arteries, leading to multifocal plaque development. The inflammatory process involves the activation of immune cells and various inflammatory pathways. Anti-inflammatory drugs have been shown to be effective in reducing cardiovascular events in individuals with coronary disease. However, their use is still limited due to concerns about long-term follow-up, cost-effectiveness, adverse effects, and the identification of the ideal patient profile to obtain maximum benefits. This review aims to improve the understanding of inflammation in coronary atherosclerosis and explore potential therapeutic interventions, encompassing both traditional and non-traditional anti-inflammatory approaches. By addressing these concepts, we seek to contribute to the advancement of knowledge about this type of treatment for coronary artery disease.

## 1. Introduction

Coronary artery disease (CAD) is an escalating chronic global disease associated with multiple adverse clinical outcomes, posing a significant threat worldwide [1,2]. Acute myocardial infarction (AMI) has an average mortality rate of 6% in those treated according to the main guidelines and, among those who survive, there is a risk of 5% per year of the recurrence of ischemic events [3,4]. CAD is primarily caused by the presence of atherosclerosis, a chronic inflammatory process resulting from an imbalance in lipid metabolism and vascular function, and a maladaptive immune response.

Most adult individuals have evidence of coronary atherosclerosis, but the occurrence of events is relatively low, as the main mechanism leading to an acute coronary syndrome (ACS) is the rupture of a coronary atherosclerotic plaque or endothelial erosion with the formation of an occluding thrombus [1]. In addition to the presence of a lipid-rich necrotic core, immune cells play a crucial role in plaque instability by promoting activation and production inflammatory cytokines [2,3] (Figure 1).

Several studies have provided evidence supporting the therapeutic potential of modulating the inflammatory response in both ACS and stable scenarios. This is particularly relevant given the significant morbidity and mortality associated with coronary disease, despite advancements in treatments targeting hypercholesterolemia, platelet aggregation, and risk factor management. The aim of this article is to review concepts pertaining to the role of inflammation in atherosclerosis and to provide an overview of recent clinical trials that have evaluated drugs with anti-inflammatory properties in CAD.

## 2. Atherosclerosis and Inflammation

Atherosclerotic lesions arise from a chronic systemic lipid immunoinflammatory disease that affects the innermost layer of the arteries and is associated with well-known cardiovascular risk factors, such as hypertension, diabetes, smoking, and hypercholesterolemia [1,2,4,5]. Traditionally, the pathogenesis of atherosclerosis has been primarily attributed to lipid accumulation within the arterial walls. Indeed, the presence of a fatty streak due to the accumulation of foam cells is an initial stage of this process. Although this alteration is frequently observed in young individuals, its subsequent course is unpredictable, as it can either remain a quiescent lesion or progress into an atheromatous plaque [4].

Myeloid cells play an important role in the development of atherosclerosis. Following recruitment through the damaged endothelium, monocytes undergo differentiation into dendritic cells and proinflammatory macrophages within the subendothelium [1,6]. Dendritic cells act as an intermediary link between the initial innate immune response and the subsequent adaptive immune response, triggered by the presentation of antigens found within atherosclerotic plaques [7]. Macrophages contribute to the progression of the disease by producing proteolytic enzymes, accumulating intracellular lipids, and secreting cytokines. The formation of a “necrotic core” observed in atheroma occurs due to the accumulation of non-phagocytosed macrophages and their debris, rendering the lesion more susceptible to rupture [4,6]. Other cells that promote plaque progression, destabilization, and thrombosis are neutrophils. They achieve this through the release of neutrophil extracellular traps, the secretion of reactive oxygen species, and their capacity to attract proinflammatory monocytes to the arterial wall [7,8].

In addition, another pathway that has been studied in recent years is the formation of the NLRP3 (NACHT, LRR, and protein-3-containing PYD domains) inflammasome in macrophages after the activation of modified lipids, leading to the production of mature IL-1β, a key step in the inflammatory signaling process [9].

T lymphocytes orchestrate an important stage of the immune response in atheroma through the Th1 response. This response is characterized by the production of cytokines such as IFN-y and TNF-a, which induce macrophage activation and the release of vasoactive factors (e.g., nitric oxide) and pro-inflammatory mediators [4]. There are other subtypes of T cells (Th2, Th17, and Treg) that participated in atherosclerosis, but their roles are less well defined [6].

Cytokines also have a relevant role in the pathogenesis of atherosclerosis. Several immune cells mentioned above, when activated, secrete inflammatory cytokines such as interferon-γ, interleukin-1β, and tumor necrosis factor-α. These cytokines, in turn, stimulate the production of substantial amounts of interleukin-6 (IL-6). IL-6 creates an environment that favors the large-scale production of acute-phase reagents by the liver, such as C-reactive protein (CRP), serum amyloid A, and fibrinogen [1,3]. Furthermore, individuals with metabolic syndrome and obesity often exhibit an increased production of “adipokines”, which can also potentialize the body’s inflammatory response [1]. This hostile inflammatory environment can have detrimental effects on atherosclerotic lesions, leading to their destabilization, plaque rupture, thrombosis, and subsequent ischemic events.

For decades, there has been significant interest in studying the use of biomarkers to assess inflammatory processes. High levels of high-sensitivity C-reactive protein (hs-CRP) and IL-6 were demonstrated to be strongly associated with vascular events [10]. However, CRP is produced by the liver and likely represents the final pathway of IL-1/IL-6 activation, so is unlikely to directly contribute to the occurrence of events such as ACS [5]. Despite this, hs-CRP has emerged as a leading biomarker and has been widely studied as an adjunct in predicting the overall risk of cardiovascular disease [11,12].

A recent study showed that in a population of patients at high risk for or with atherosclerosis, hs-CRP was a stronger predictor of cardiovascular events, cardiovascular death, and all-cause death than residual cholesterol risk (as measured by low-density lipoprotein cholesterol) [13]. This highlights the potential usefulness of CRP as a prognostic tool in assessing cardiovascular risk in CAD. However, it is important to note that while measurable circulating biomarkers such as CRP provide valuable information, they are inherently of limited use regarding information about the location and extent of atherosclerosis.

## 3. Anti-Inflammatory Therapy for Coronary Artery Disease

There are numerous firmly established therapeutic strategies to enhance secondary prevention in atherosclerosis. These include the use of statins, antiplatelet agents, smoking cessation initiatives, and lifestyle interventions [14,15]. However, it is noteworthy that coronary heart disease continues to be the leading global cause of mortality. Consequently, anti-inflammatory therapies have emerged as promising candidates for decreasing residual cardiovascular risk [13]. Although inflammation plays a critical role in CAD, a significant challenge lies in comprehending the central mechanisms of pathogenesis and identifying the specific inflammatory pathways responsible for disease progression.

Numerous systemic anti-inflammatory agents have been well studied, such as glucocorticoids, non-steroidal anti-inflammatory drugs, and anti-cytokine agents. However, their long-term use as therapeutic options in CAD to modulate atherosclerosis is hindered by their associated undesirable side effects. These limitations make them less favorable candidates for prolonged therapy. Several alternative pathways have been under consideration for further investigation, including the leukotrienes, which function as inflammatory eicosanoid mediators originating from leukocytes via the oxidation of arachidonic acid and eicosapentaenoic acid. Intriguingly, specific antileukotriene agents commonly employed in asthma treatment, including montelukast, zafirlukast, pranlukast, and pobilucast, have demonstrated the capacity to mitigate cardiac events in animal models and in small observational clinical trials involving human participants [16,17].

Therefore, in recent years numerous clinical trials have been conducted to explore the efficacy of various drugs in modulating inflammation in atherosclerosis with promising clinical outcomes, including a reduction in cardiac events and/or decrease in inflammatory markers in patients with CAD [18,19,20].

In the next paragraphs, this document presents a comprehensive summary of available evidence derived from randomized clinical trials that evaluated drugs with anti-inflammatory properties in patients with coronary artery disease and reported clinical outcomes and/or changes in inflammation biomarkers following the use of these therapies. For simple classification purposes, we divided these studies into three clusters according to the type of drug used: colchicine (Table 1), other anti-inflammatories (Table 2), and non-traditional therapies with anti-inflammatory properties (Table 3).

## 4. Relevant Clinical Trials That Have Used Colchicine in CAD

Colchicine has been used for a variety of inflammatory conditions over the years due to its action in binding to tubulin and inhibiting its polymerization, with the subsequent disruption of cellular cytoskeleton, mitosis, and intracellular transport activities [35]. In the last few years, several trials have demonstrated that colchicine reduces major adverse cardiovascular events in patients with coronary disease. Considering its wide availability, low cost, safety, and favorable side-effect profile, this drug has emerged as a potential oral cardiovascular treatment in CAD [36]. It has recently received approval from the FDA (Food and Drug Administration) for use in a low-dose regimen, in line with recent guidelines recommending the consideration of its use in this population [14,15].

In the setting of atherosclerosis, colchicine has been demonstrated to not only act by reducing the migration, adhesion, and activation of neutrophils to the inflamed endothelium, but also suppress the assembly and activation of the NLRP3 and reduce the inflammatory cytokines that play a relevant role in plaque development and instability [37]. Martinez et al. demonstrated that the use of oral colchicine, when compared to placebo, 6 to 24 h before cardiac catheterization, was related to a significant reduction in interleukins (IL-1B, IL-6, and IL-18) measured directly in the coronary artery bed in patients with ACS [22]. This finding shows the importance of using an anti-inflammatory drug in acute coronary scenarios, since pro-inflammatory cytokines amplify the inflammatory response and increase the chances of plaque instability in non-culprit lesions.

Another study tested the use of colchicine in higher doses (1.2 mg followed by 0.6 mg after 1 h) in patients with CAD who underwent elective coronary angioplasty, without finding a reduction in the incidence of myocardial injury or MACE outcomes (any death, non-fatal AMI, or target vessel revascularization) within 30 days of its use, but the medication was associated with an attenuation in CRP and IL-6 values within up to 24 h [26]. On the other hand, an open, single-center study evaluated the use of colchicine (1 mg daily) in 44 patients with STEMI after angioplasty and found no difference in peak CRP measured during hospitalization or in clinical outcomes at 30 days [23].

Two randomized trials tested the use of colchicine in patients after a diagnosis of acute coronary syndrome. The COPS trial failed to show any benefit with the use of low-dose colchicine in reducing all-cause death, ACS, urgent revascularization caused by ischemia, and stroke when compared to placebo in 795 individuals [24]. On the other hand, Tardif et al. showed that using once-daily colchicine in 4745 patients after an ACS was associated with a reduction in the combined endpoint of cardiovascular death, resuscitated cardiac arrest, stroke, AMI, and unplanned revascularization (HR of 0.77) within 20 months [18].

Colchicine has also been evaluated in chronic coronary disease. In 2013, Nidorf et al. performed a prospective, single-center, randomized study that evaluated the addition of colchicine to statins and other standard secondary prevention therapies in stable coronary disease and observed a relevant reduction in ACS, out-of-hospital arrest, and ischemic stroke for a median of 3 years. However, the significance of these results was uncertain due to their open and single-blind design [21]. Years later, Lodoco2 was an international multicenter study that included 5572 patients with chronic coronary disease who used 0.5 mg of colchicine once a day for 28 months. The intervention was associated with a 2.8% absolute reduction in the primary endpoint when compared to placebo (HR 0.69, 95% CI 0.57–0.83, *p* < 0.001) [25].

In summary, there is substantial evidence supporting the potential use of low-dose colchicine as a therapy for CAD. However, despite all these published trials, there are lingering uncertainties regarding its optimal use and long-term implications. It is important to consider that colchicine is not without risks, as approximately 20% of patients may present gastrointestinal symptoms at some point. Moreover, certain studies have indicated an increase in non-cardiac-related deaths among individuals undergoing this treatment, with no clear explanation provided to date. There is a suspicion that this increase in mortality may be associated with a high risk of fatal infections [24,36]. Therefore, an extended follow-up period is necessary to comprehensively evaluate its safety profile and overall clinical benefits.

Besides this, identifying the ideal target population for this therapy is a crucial consideration. Despite the numerous treatments for CAD, it is important to focus on individuals with high residual risk, as they are most likely to benefit from colchicine. However, the tools for accurately identifying them are not yet well-established. As mentioned above, some studies did not achieve the objective of reducing inflammatory markers with this therapy. It is possible that colchicine could still have a positive impact on clinical outcomes independently of these markers. Notably, neither the LoDoCo nor COLCOT studies specifically selected patients based on elevated inflammatory risk, which is currently defined as having high-sensitivity CRP > 2 mg/L. Therefore, implementing a specific biomarker strategy could help identify a subset of target patients, enhancing the benefit-to-risk ratio of therapy.

## 5. Relevant Clinical Trials That Have Used Other Anti-Inflammatories Drugs in CAD

Cytokines play a crucial role in mediating inflammation and immune responses. Interleukyne-1β (IL-1β) is a key cytokine involved in the inflammatory cascade and has been shown to upregulate the production of IL-6, which is known to be associated with the activation of pathways linking inflammation to vascular events [1,10].

Smaller trials with limited statistical power have evaluated IL-1α and IL-1β inhibition using the anakinra receptor antagonist [27,30]. The MRC-ILA study included patients following a non-ST elevation myocardial infarction (NSTEMI) who received either anakinra or a placebo and demonstrated a decrease in the area under curve (AUC) of hs-PCR after 14 days of treatment when compared to the placebo group. Abbate et al. showed in a population with STEMI a decrease in the AUC of CRP accompanied by significant reductions in deaths or new-onset or worsening heart failure with anakinra. Another trial tested tocilizumab, an anti-IL-6R antibody, in patients with NSTEMI and observed an attenuated inflammatory response, a reduction in coronary angioplasty-related troponin release, and a favorable safety profile [28].

Canakinumab is a fully human monoclonal antibody that specifically targets IL-1β, inhibiting the NLRP3 to IL-1 to IL-6 pathway of innate immunity. It was evaluated in two different doses in the large-scale CANTOS trial. The trial included individuals with CAD who had a previous myocardial infarction and us-CRP > 2 mg/L, despite receiving optimized medical therapy. After three years, this drug effectively reduced cardiovascular events (AMI, stroke, and death from cardiovascular causes) compared to placebo, although only at one dose and not in terms of single endpoints [19]. The extent of risk reduction for cardiac events was greatest among patients with the largest reductions in their levels of IL-6 and hs-CRP. Notably, these favorable results came with an increase in infections, including fatal infections.

In contrast, Ridker et al. tested low-dose methotrexate, a purine metabolism inhibitor, in individuals with a history of AMI or multivessel CAD who also had diabetes or metabolic syndrome and failed to demonstrate a beneficial effect on cardiac events or a reduction in CRP, IL-6, and IL-1b levels over two years [29].

It is important to note that the results of the CANTOS trial represent a cornerstone in our understanding of treating inflammation in CAD. They provided a proof-of-concept that inhibiting inflammation can prevent atherosclerosis-related events in humans. The divergent results between the CANTOS and CIRT trials indicate the need for further research to better understand the complex nature of inflammation in CAD, and that anti-inflammatory therapies for coronary atherosclerosis should be aimed at specific pathways. The IL-1β-IL-6-CRP pathway has been demonstrated so far to be a promising target. However, it is important to consider the cost-effectiveness of using human antibodies in this population and their immune-event side effects.

## 6. Relevant Clinical Trials That Have Used Non-Traditional Therapies with Anti-Inflammatories Properties in CAD

A considerable number of patients use non-traditional products, such as herbal medicines and dietary supplements, for the treatment of various diseases, often including substances with known anti-inflammatory properties. These bio-based products cover a wide range of options, including plant-based products, animal-derived extracts, vitamins, minerals, fatty acids, amino acids, proteins, probiotics, and prebiotics [38]. Although limited in scale, several of these non-traditional therapies have been investigated in small-scale clinical trials in the context of coronary disease.

Kapoor et al. conducted a small study to evaluate the effects of *Terminalia arjuana*, a plant-based product known for its anti-inflammatory and antioxidant properties, compared to placebo in 116 individuals with stable coronary disease who were receiving optimized treatment for 6 months. Its use was associated with a decrease in triglycerides, very-low-dense-lipoprotein (VLDL), us-CRP, IL-18, IL-6, and TNF-alpha. However, despite these favorable changes in biomarkers, there was no difference observed in clinical events such as ACS, AMI, stroke, coronary revascularization, and cardiac deaths [31]. Although several trials have shown the efficacy of Arjuana in a wide range of cardiovascular disease, there are still considerations that need to be addressed to ensure its safe and effective use, including the standardization of its extraction, the investigation of potential drug interactions, and the performance of larger trials with a more diverse population [39].

L-carnitine, a natural compound produced by the body and present in certain foods, is a vitamin-like substance that plays a role in increasing energy production within mitochondria and was recently demonstrated to modulate inflammatory cells. In a clinical trial involving patients with CAD who were not taking statins, the effects of L-carnitine were compared to a placebo. Over a 12-week period, the study found a significant decrease in the levels of CRP, IL-6, and TNF-alpha among those who received L-carnitine compared to the placebo group (*p* < 0.03) [32].

In a study conducted by Antunina et al., the effects of alpha-lipoic acid (ALA), a natural compound known for its antioxidant and anti-inflammatory properties, were evaluated in 112 patients who suffered from AMI and had diabetes. The study aimed to evaluate the impact of ALA supplementation on inflammatory markers. After a 4-month period, researchers observed that ALA supplementation was associated with a significant decrease in the levels of CRP, IL-6, and TNF-alpha when compared to the placebo group [33].

The Guanxin Danshen Dropping pill is a Chinese herbal product that can be used isolated (ABC group) or in combination with andrographis (ABCD group) in traditional Chinese medicine. Its precise mechanism of action is not fully understood, but it is believed to improve blood flow, reduce inflammation, protect against oxidative stress, and inhibit platelet aggregation. It is commonly used for angina, coronary artery disease, and myocardial infarction [34]. Xiao-Juan et al. performed a randomized clinical trial comparing the ABC group to the ABCD group among subjects undergoing coronary angioplasty for unstable angina. The pill in the ABCD group exhibited a significant reduction in us-CRP, IL-6, and TNF-alpha levels. Furthermore, patients in the ABCD group experienced greater relief from angina symptoms [40].

The use of alternative therapies is widespread among different populations and ethnic groups, generating great interest in better understanding their pharmacological properties and testing them in different scenarios. In recent years, several non-traditional substances have undergone preclinical trials for CAD with promising results, as cited above. However, the effectiveness of these interventions has been assessed with limited or no high-quality evidence. Therefore, there is an urgent need for additional mechanistic and translational studies, as well as well-designed and adequately powered randomized controlled trials, to establish guidelines and recommendations.

## 7. Future Perspectives

Brazilian green propolis extract (EPP-AF^®^) is a natural resin derived from various bioactive plant sources, produced by *Apis mellifera* bees. This extract has antioxidant, anti-inflammatory, and immunomodulatory properties [41,42]. Previous studies have demonstrated the potential of propolis in reducing inflammatory biomarkers and symptoms of inflammatory diseases in different contexts [42,43,44].

For instance, in a population with chronic kidney disease undergoing hemodialysis, the use of propolis for four weeks led to significant reductions in the levels of TNF-alpha, gamma interferon, IL-1, and other cytokines [45]. Additionally, a randomized study involving patients hospitalized with COVID-19 revealed that those who received daily doses of 400 to 800 mg of propolis for seven days had a shorter hospital stay compared to those who received placebo [46]. These data indicate the safety of the dosage in randomized trials, providing opportunities for its utilization in further clinical studies.

The anti-inflammatory and immunomodulatory capacity of propolis observed in these clinical studies involving interleukins offers potential for its use in the context of coronary disease, where inflammation plays a crucial role in the development of lesions. A recent review highlighted the cardiovascular effects of green propolis, emphasizing its role in endothelial and myocardial protection, as well as its antiangiogenic properties [47]. Furthermore, another study examined several samples of propolis and revealed its potential to influence in vitro platelet aggregation function tests, suggesting that this natural product may have antiaggregant properties [48].

Our team is currently in the recruitment phase of a randomized, placebo-controlled, double-blind, single-center pilot clinical trial designed to evaluate the efficacy of propolis in patients with stable coronary artery disease who are receiving optimal clinical treatment but continue to experience angina symptoms. The primary objective of the trial is to assess whether the use of this substance provides benefits in terms of improved functional capacity in stress tests and/or a reduction in hs-CRP levels after six weeks of treatment.

Furthermore, the success of IL-1β targeting highlights the inflammasome pathway as a promising target for other therapeutic interventions. For example, the NLRP3 inflammasome, along with the downstream cytokines IL-1β, IL-18, and IL-6, is an attractive candidate target for future drugs. An ongoing clinical trial (NCT05021835) is currently evaluating the use of an IL-6 inhibitor (ziltivekimab) versus a placebo among 6000 patients with chronic kidney disease and elevated hs-CRP. The study is aiming to evaluate whether this drug reduces the rates of cardiovascular events in this population. The LILACS study (NCT03113773) is a phase I/II study that was recently completed and evaluated the safety of low-dose IL-2 in cardiovascular patients. The aim of this study was to investigate whether this approach can increase the number and function of regulatory T cells (Tregs) and ultimately promote plaque stabilization and myocardial healing.

## 8. Conclusions

Chronic inflammation plays a significant role in the development of atherosclerosis and contributes to an increased risk of cardiovascular events. Targeting anti-inflammatory therapies has emerged as a promising strategy, given the growing evidence of the impact of vascular inflammation on the progression of coronary disease despite its current medical treatment. The positive effects seen with canakinumab and colchicine have spurred numerous studies to further explore this new therapeutic approach. However, current guidelines still provide a weak recommendation for its use, mainly due to uncertainties regarding long-term effects and the balance between clinical benefits and possible side effects.

Perhaps the answer lies in exploring other types of anti-inflammatory agents that have not yet been extensively tested in larger trials, such as natural substances with known anti-inflammatory properties used in non-traditional but ancient medicine. While addressing the inflammatory component of atherosclerosis is clinically relevant and leads to better clinical outcomes, more research is needed to determine the optimal drug and patient profile for maximum benefit.

## Figures and Tables

**Figure 1 pharmaceuticals-16-01242-f001:**
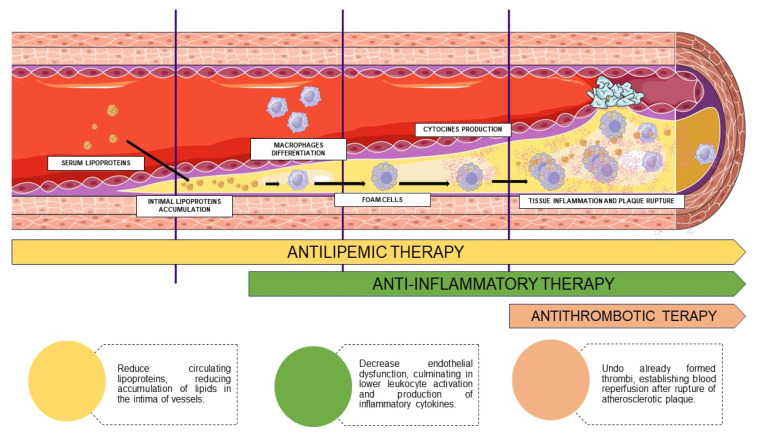
Pathophysiology of atherosclerosis associated with drug therapy and its stages—the formation of atheromatous plaque initially depends on the accumulation of lipids in the intima of the vessels; therefore, antilipemic therapy contributes to the reduction or cessation of this stimulus. However, after oxidation and the phagocytosis of these molecules by macrophages, an intense inflammatory cascade occurs, causing a cycle of the recruitment of inflammatory cells and the release of cytokines, culminating in tissue damage, allowing therapeutic targets to be used in anti-inflammatory therapy. Finally, when the plaque ruptures, there is the possibility of using thrombolytics to minimize the repercussions on the blood flow caused by the obstruction resulting from the consequent formation of thrombi.

**Table 1 pharmaceuticals-16-01242-t001:** Randomized clinical trials that have tested the use of colchicine in coronary artery disease.

Trial	Year of Study	Nº of Patients	Inclusion Criteria	Investigated Drug	Primary Outcomes	Results
Nidorf et al. * [21]	2013	532	Patients with chronic coronary disease using statins and standard secondary prevention therapy	Colchicine 0.5 mg o.d. vs. no colchicine	ACS, out-of-hospital cardiac arrest, and/or non-cardioembolic ischemic stroke	After a median of 36 months, 5.3% of patients in the colchicine group had a primary outcome versus 15% in the no colchicine group (HR 0.33, 95% CI 0.18–0.59; *p* < 0.001).
Martínez et al. [22]	2015	83	Patients with ACS (excluding STEMI), stable CAD, and controls (no evidence of lesions) who would undergo coronary angiography	Colchicine 1 mg o.d. vs. placebo	Assessment of IL-1β, IL-6, and IL-18 levels in coronary arteries	Colchicine significantly reduced the transcoronary gradients of these markers in patients with ACS (*p* = 0.028, 0.032, and 0.032, for IL-1b, IL-18, and IL-6, respectively). There was no difference for those with stable CAD.
Akodad et al. [23]	2017	44	Patients with STEMI who underwent an angioplasty of the culprit lesion	Colchicine 1 mg o.d. + OMT vs. OMT	CRP peak assessment during hospitalization	No significant difference in CRP peak between those who used or did not use colchicine (29.03 mg/L × 21.86 mg/L, respectively, *p* = 0.86).
Tardif et al. [18]	2019	4745	Patients who had AMI within 30 days and were treated with PCI and were using statins and OMT	Colchicine 0.5 mg o.d. vs. placebo	Cardiovascular death, CPR, AMI, stroke, or hospitalization for angina with subsequent revascularization	Colchicine reduced the primary endpoint (HR 0.77, 95% CI 0.61–0.96, *p* = 0.02) at 19.6 months of follow-up.
Tong et al. [24]	2020	795	Patients with ACS and evidence of CAD on coronary angiography who were treated with PCI or medical therapy	Colchicine 0.5 mg b.i.d. for 30 days followed by 0.5 mg o.d. vs. placebo	Death for any cause, ACS, ischemia-driven urgent revascularization, and stroke	Over the 12-month follow-up period, there was no significant difference in primary endpoint between the two groups (24 (6.1%) in the colchicine group versus 38 (9.5%) in the placebo group).
Nidorf et al. [25]	2020	5572	Patients with stable CAD receiving OMT for at least six months (catheterization or coronary CT angiography with significant lesion or calcium score with Agatston score > 400)	Colchicine 0.5 mg o.d. vs. placebo	Cardiovascular death, spontaneous AMI, stroke, and coronary revascularization	After 28.6 months, colchicine reduced the primary endpoint (HR 0.69, 95% CI 0.57–0.83, *p* < 0.001).
Shah et al. [26]	2020	400	Patients with stable CAD who would undergo coronary angioplasty	Colchicine 1.8 mg o.d. vs. placebo	Evaluation of myocardial injury reduction after angioplasty	No difference in myocardial injury related to angioplasty. No differences in the secondary endpoint of MACE reduction at 30 days. There was an attenuation in the increase in IL-6 and CRP in the colchicine group within 24 h after the procedure.

ACS: acute coronary syndrome; AMI: acute myocardial infarction; CAD: coronary artery disease; CPR: cardiopulmonary resuscitation; CRP: C-reactive protein; MACE: major adverse cardiac event; PCI: percutaneous coronary intervention; OMT: optimized medical treatment; STEMI: ST elevation myocardial infarction. * This study was an open-label trial with blinded endpoint adjudication (a PROBE design).

**Table 2 pharmaceuticals-16-01242-t002:** Randomized trials that have tested other anti-inflammatory drugs in coronary artery disease.

Trial	Year of Study	Nº of Patients	Inclusion Criteria	Investigated Drug	Primary Outcomes	Results
Morton et al. [27]	2015	182	Patients with NSTEMI presenting with <48 h from onset of chest pain	Anakinra (100 mg o.d.) vs. placebo	Evaluation of AUC of serum hs-CRP over the first 7 days	The hs-CRP AUC was 21.98 mg day/L in the IL-1ra group and 43.50 mg day/L in the placebo group, with the geometric mean ratio between IL-1ra and placebo being 0.51 (95% CI: 0.32–0.79, *p* = 0.0028).
Kleveland et al. [28]	2016	117	Patients with NSTEMI scheduled for coronary angiography.	Tocilizumab (single dose of 280 mg) or placebo	Evaluation of AUC of serum hs-CRP on days 1 and 3	The median AUC for hs-CRP during hospitalization was 2.1 times greater in the placebo group than in the tocilizumab group (4.2 vs. 2.0 mg/L/h, *p* < 0.001).
Ridker et al. [19]	2017	10061	Patients with previous AMI and high hs-CRP (>2 mg/L)	Canakinumab (50 mg, 150 mg e 300 mg) vs. placebo	Nonfatal AMI, nonfatal stroke, and cardiovascular death	Only the 150 mg dose showed a significant reduction in the primary outcome vs. placebo (HR 0.85, *p* = 0.02) after 3.7 years. There was a reduction in hs-CRP and IL-6 levels with all doses.
Ridker et al. [29]	2019	4786	Patients with a history of AMI or multivessel CAD and T2DM or metabolic syndrome	Low-dose methotrexate vs. placebo	Nonfatal AMI, nonfatal stroke, and cardiovascular death	No difference in primary outcome between groups (HR 0.96, CI 0.79–1.16 *p* = 0.67). No reductions in hsCRP, IL-6, and IL-1β levels were observed.
Abbate et al. [30]	2020	99	Patients with an STEMI who underwent urgent coronary angiography within 12 h of symptom onset	Anakinra 100 mg o.d. vs. Anakinra 100 mg b.i.d. vs. placebo	Evaluation of AUC of serum hs-CRP at baseline, 72 h, and day 14	The AUC of hs-CRP was lower in the anakinra group versus the placebo (median, 67 (IQ range 39–120) versus 214 (IQ range 131–394) mg day/L; *p* < 0.001. No significant differences were observed between the two regimens of anakinra. The incidence of death or new-onset or worsening HF was lower with anakinra (*p* = 0.046).

AMI: acute myocardial infarction; AUC: area under the curve; CAD: coronary artery disease; HF: heart failure; hs-CRP: high-sensitivity C-reactive protein; IL-1ra: interleukin 1 receptor antagonist; IQ: interquartile; NSTEMI: non-ST elevation myocardial infarction; STEMI: ST elevation myocardial infarction; T2DM: type 2 diabetes mellitus.

**Table 3 pharmaceuticals-16-01242-t003:** Randomized clinical trials that have tested drugs with anti-inflammatory properties in coronary artery disease.

Trial	Year of Study	Nº of Patients	Inclusion Criteria	Investigated Drug	Primary Outcomes	Results
Kapoor et al. [31]	2015	116	Patients with stable CAD and OMT	Terminalia arjuna 500 mg b.i.d. vs. placebo	Assessment of levels of inflammatory markers (IL-6, IL-18, TNF-α, IL-10, and hs-CRP) and lipid profile	T. arjuna reduced CT, TG, VLDL, IL-6, IL-18, TNF-α, and hsCRP. There was an increase in HDL and IL-10. No differences were observed in MACE.
Bor-Jen Lee et al. [32]	2015	47	Patients with CAD (catheterization with lesion > 50% or previous angioplasty)	L-carnitine 1000 mg o.d. vs. placebo	Assessment of levels of inflammatory markers (IL-6, TNF-α, and CRP)	LC supplementation reduced CRP levels by 10%, IL-6 by 17%, and TNF-α by 6% (*p* = 0.03) when compared to placebo after 12 weeks.
Altunina et al. [33]	2020	112	Patients with T2DM and a history of previous AMI using oral hypoglycemic agents, antiplatelet agents, and statins	ALA 600 mg vs. placebo	Assessment of CRP, IL-6, TNF-α, and IL-10 levels	ALA was associated with a 30.9% reduction in CRP, 29.7% in IL-6, and 22.7% in TNF-α after 4 months.
Wang et al. [34]	2021	154	Patients aged 40–75 years with unstable angina who underwent angioplasty within the last 48 h	ABCD group (Guanxin Danshen Dropping Pill 0.4 g and andrographis 0.2 g) vs. ABC (Guanxin Danshen Dropping Pill 0.4 g)	hs-CRP evaluation	After 30 days, the ABCD group showed a reduction in hs-CRP compared to the ABC group (2.96 mg/L × 1.54 mg/L, *p* < 0.05). The ABCD group also showed a reduction in IL-6 and TNF-α levels. There was an improvement in the angina score in the ABCD group versus the ABC group (*p* < 0.05).

ALA: alpha lipoic acid; AMI: acute myocardial infarction; CAD: coronary artery disease; CRP: C-reactive protein; HDL: high-density lipoprotein; hs-CRP: high sensitivity C-reactive protein; LC: L-carnitine; OMT: optimized medical treatment; T2DM: type 2 diabetes mellitus; TC: total cholesterol; TG: triglycerides; VLDL: very-low-density lipoprotein.

## Data Availability

Data sharing is not applicable.

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
