# Peer review of "Inflammation in Coronary Atherosclerosis: Insights into Pathogenesis and Therapeutic Potential of Anti-Inflammatory Drugs"

_pharmaceuticals, 2023, doi:10.3390/ph16091242_

Round 1

Reviewer 1 Report

There are already overwhelming redundant articles related to this topic in the literature (Few examples include

https://pubmed.ncbi.nlm.nih.gov/33170943/

https://www.ncbi.nlm.nih.gov/pmc/articles/PMC10225146/

https://www.mdpi.com/2075-1729/13/8/1669

https://www.sciencedirect.com/science/article/pii/S2666667720301306

https://www.ncbi.nlm.nih.gov/pmc/articles/PMC7882495/

It is not clear what are the new insights that review will really shed light on and why the readers will prefer it over existing articles?

Author Response

Greetings,

I agree that the subject of "atherosclerosis and inflammation" is of significant contemporary interest, particularly in light of recent publications such as the CANTOS trial and other clinical trials involving colchicine. These studies have provided compelling evidence on the merits of targeting inflammation as a therapeutic approach in the context of atherosclerotic disease. A central objective of our research was to review the main studies on the use of colchicine and immunobiological agents and to bring them together comparing different clinical scenarios, immediate post-AMI, longer follow-up after ACS and stable CAD. Notably, our team performed an exhaustive review of the topic and assembled these works in an as yet unpublished form.

Furthermore, it is worth noting that the choice of the anti-inflammatory agent in CAD cenarios remains an ongoing topic of discussion, primarily due to cost-effectiveness considerations.  Currently, colchicine stands out as the predominant affordable therapeutic option (having recently been approved by the FDA for this purpose). However, our attention was extended to the so-called "nontraditional therapies" with anti-inflammatory properties that were studied in the context of coronary artery disease respecting the methodology of a randomized clinical trial and that had description of clinical outcome and/or markers of inflammation. To date, we have not found a review of more traditional databases that pooled the evidence from these smaller clinical trials as proposed in the current article.

The main idea was to evaluate the possibility of seeking new alternatives for the treatment of the inflammatory pathway of CAD using more accessible therapies, with a good safety profile and that have demonstrated efficacy in these smaller clinical trials. Thus, our intention was to demonstrate that it might be feasible to submit these alternative drugs to larger clinical trials, as we are doing with the use of green propolis extract in our institution.

Thank you for your attention and considerations. 

Reviewer 2 Report

In this manuscript, Clara Salles Figueiredo and collaborators deal with a complex, very interesting and current topic. The reading is pleasant and linear as the authors have explained the topic in a practical and concise way, also aided by the tables and the figure which favor the reader's understanding. However, I must point out some weaknesses that reduce its appeal and novelty and which, in my opinion, must necessarily be answered.

1) On page 2 (lines 65-67) The review does not mention other cells as neutrophils and dendritic cells and platelets, whose role in the pathophysiology of atherosclerosis, although known for a long time, has had recent developments above all due to the growing attention towards the link between atherosclerosis and hematopoiesis dysfunction. The interest in neutrophils (and platelets) has practical implications, there are growing reports in the literature on the usefulness of some parameters derived from the blood count in CV risk stratification such as the neutrophil/lymphocyte ratio or the platelet X neutrophil/lymphocyte ratio and so on. I suggest the following references which can help authors to integrate information:

Tucker B, Ephraums J, King TW, Abburi K, Rye KA, Cochran BJ. Impact of Impaired Cholesterol Homeostasis on Neutrophils in Atherosclerosis. Arterioscler Thromb Vasc Biol. 2023 May;43(5):618-627. doi: 10.1161/ATVBAHA.123.316246. Epub 2023 Mar 23. PMID: 36951066.

Silvestre-Roig C, Braster Q, Ortega-Gomez A, Soehnlein O. Neutrophils as regulators of cardiovascular inflammation. Nat Rev Cardiol. 2020 Jun;17(6):327-340. doi: 10.1038/s41569-019-0326-7. Epub 2020 Jan 29. PMID: 31996800.

Libby P. Inflammation during the life cycle of the atherosclerotic plaque. Cardiovasc Res. 2021 Nov 22;117(13):2525-2536. doi: 10.1093/cvr/cvab303. PMID: 34550337; PMCID: PMC8783385.

Engelen SE, Robinson AJB, Zurke YX, Monaco C. Therapeutic strategies targeting inflammation and immunity in atherosclerosis: how to proceed? Nat Rev Cardiol. 2022 Aug;19(8):522-542. doi: 10.1038/s41569-021-00668-4. Epub 2022 Jan 31. PMID: 35102320; PMCID: PMC8802279.

2) on page 3 (line 246 and following) "Relevant clinical trials that used non-traditional therapies with anti-inflammatories properties in CAD"

It would be better to specify that this therapeutic approach derives from the observation of food style as it is now evident that diet correlates with the degree of systemic inflammation as well as the lipid and glucid profile. Furthermore, always in this regard it would be useful to mention the role of the microbiota, a topic that in recent years has been drawing attention in the field of CV prevention to the potential modulation role of inflammation.

Karam G, Agarwal A, Sadeghirad B, Jalink M, Hitchcock CL, Ge L, Kiflen R, Ahmed W, Zea AM, Milenkovic J, Chedrawe MA, Rabassa M, El Dib R, Goldenberg JZ, Guyatt GH, Boyce E, Johnston BC. Comparison of seven popular structured dietary programmes and risk of mortality and major cardiovascular events in patients at increased cardiovascular risk: systematic review and network meta-analysis. BMJ. 2023 Mar 29;380:e072003. doi: 10.1136/bmj-2022-072003. PMID: 36990505; PMCID: PMC10053756.

Witkowski M, Weeks TL, Hazen SL. Gut Microbiota and Cardiovascular Disease. Circ Res. 2020 Jul 31;127(4):553-570. doi: 10.1161/CIRCRESAHA.120.316242. Epub 2020 Jul 30. PMID: 32762536; PMCID: PMC7416843.

Brecht P, Dring JC, Yanez F, Styczeń A, Mertowska P, Mertowski S, Grywalska E. How Do Minerals, Vitamins, and Intestinal Microbiota Affect the Development and Progression of Heart Disease in Adult and Pediatric Patients? Nutrients. 2023 Jul 24;15(14):3264. doi: 10.3390/nu15143264. PMID: 37513682; PMCID: PMC10384570.

3 Conclusion. I personally believe that there is no pharmacological "magic bullet" for the prevention of inflammation, I take the liberty of advising the authors to highlight the role of lifestyle and non-pharmacologic factors (exercise, weight control, stress and so on) as recently reinforced in the 2023 AHA/ACC guidelines on chronic ischemic heart disease

Author Response

Greetings,

I sincerely appreciate your comments and contributions.

Comment 1:

I agree with the reviewer's perspective on the significance of referencing the roles of other cells within the innate immune response, including dendritic cells and neutrophils. Upon reviewing the provided references, we engaged in a discussion recognizing the need to emphasize these cells in the context of pathophysiology, as they could potentially offer viable therapeutic targets for atherosclerosis (as indicated in lines 90-99). I extend my gratitude for the recommended texts; they were enlightening and aided in comprehending the profound importance of the subject matter.

Comment 2:

We acknowledge the pivotal role that dietary components play in the development of various diseases. Additionally, there is a noticeable surge in interest surrounding the investigation of microbiota in diverse clinical contexts, including coronary artery disease. The extensive array of articles accessible on PubMed, including several clinical trials, is indeed impressive.

However, while recognizing the significance of these topics, it is essential to reiterate that the primary focus of our article revolves around the review of inflammatory drugs in coronary artery disease. Our intention is to provide comprehensive insights into randomized clinical trials that have tested anti-inflammatories drugs and non-traditional drugs possessing anti-inflammatory properties. We are cautious about diverging from our core objective, which involves discussing established drug therapies that have undergone scrutiny, as well as highlighting potential candidates for more extensive clinical trials.

Comment 3:

I fully agree that the recognition of healthy behaviors is essential in the prevention of secondary CAD and should be constantly highlighted. In fact, we have included a reference to the importance of this topic in our article, as indicated in lines 132-134.

Again, I genuinely value your review and trust that we have met some of your requirements.

Yours sincerely,Parte superior do formulário

Clara

Reviewer 3 Report

Dear Authors,

The review raises important and still topical issues of inflammation in atherosclerosis, especially in the aspect of its inhibition and new therapeutical pathways development. 

The submitted paper is well-written and comprehensively describes the current state of knowledge. The only issue is the lack of any information of eicosanoids and their role in atherosclerosis, especially leukotrienes. (i.ex.:10.5603/AA.2022.0013) This should not be committed, considering that there are studies showing the beneficial impact of antileukotriene drugs on cardiological outcomes. (10.1016/j.jaci.2011.11.052). Also thromboxanes are believed to be related to occurrence of adverse cardiovascular events after endovascular treatment. (https://pubmed.ncbi.nlm.nih.gov/27511998/)

The provided links are only the samples and suggestions, but the subject of eicosanoids in this article should not be completely omited.

Kind regards

Minor spelling check suggested. 

Author Response

Greetings,

I would like to express gratitude for your comments and contributions. We had not previously considered including information regarding the eicosanoid pathway and its participation in the atherosclerosis pathogenesis, and your comment was important to new considerations. 

I found the studies conducted with leukotriene inhibitors quite intriguing, especially one recently published in this journal (https://doi.org/10.3390/ph15091147). It's noteworthy that several of these studies have shown promising outcomes in animal models or cohort studies, as demonstrated in a meta-analysis published in 2018 (10.1016/j.biopha.2018.07.033). Unfortunately, I didn't come across many human studies. Ingelsson et al. conducted a well-designed retrospective study on a nationwide population-based cohort of around 7 million Swedish individuals. They hypothesized a potential role for montelukast in the secondary prevention of cardiovascular diseases (CVDs) and exhibited a reduced risk of recurrent stroke in patients exposed to the drug (HR, 0.62; 95% CI, 0.38–0.99), along with a lower risk of recurrent MI in male subjects (HR, 0.65; 95% CI, 0.43–0.99).

As described in our methodology, the intention of this review was solely to reference studies with randomized clinical trial methodology. Therefore, I don't find it pertinent to include this study. If we were to consider the various analyses of prospective and retrospective cohorts on the use of non-traditional therapies in the context of CAD, we would have to incorporate a significantly larger number of studies. In a comprehensive PubMed search, I came across a clinical study (10.1378/chest.07-0831) that assessed whether the use of montelukast and theophylline could be associated with a reduction in inflammatory markers involved in cardiovascular diseases. They observed a decrease in CRP levels and serum lipid levels. After deliberation among the authors, we decided not to include this study in our review since the studied population consisted of asthmatic individuals, and the use of this medication as a secondary prevention therapy in CAD, which is the focus of our discussion, was not considered.

I genuinely believe that conducting additional clinical trials is imperative to substantiate this hypothesis.

Nevertheless, given the significance of the topic, we deem that the leukotriene pathway should be mentioned as a potential therapeutic target in CAD, and we have incorporated this information into the text (lines 145-151). Once again, I sincerely appreciate your review and hope to have addressed some of your requests.

Best regards,

Clara

Round 2

Reviewer 1 Report

Accept in present form

Reviewer 2 Report

Compliments. I have no other comments. 

Reviewer 3 Report

Dear Authors,

Thank you for the improvements. I see your point of view and respect it. I do not have any further questions or requests.

Kind regards